# Targeting the IL-6/STAT3 Signalling Cascade to Reverse Tamoxifen Resistance in Estrogen Receptor Positive Breast Cancer

**DOI:** 10.3390/cancers13071511

**Published:** 2021-03-25

**Authors:** Ho Tsoi, Ellen P. S. Man, Ka Man Chau, Ui-Soon Khoo

**Affiliations:** Department of Pathology, Li Ka Shing Faculty of Medicine, The University of Hong Kong, Hong Kong, China; tsoiho@hku.hk (H.T.); ellenman@hku.hk (E.P.S.M.); 15435717@life.hkbu.edu.hk (K.M.C.)

**Keywords:** breast cancer, interleukin-6, interleukin-6 receptor, tamoxifen resistance, Tocilizumab, STAT3

## Abstract

**Simple Summary:**

This study identifies the molecular mechanisms through which BQ323636.1 can enhance IL-6 and IL-6R expression, which leads to the activation of STAT3 and the development of tamoxifen resistance in ER+ breast cancer. We demonstrated a statistically significant association of IL-6R with tamoxifen resistance; patients with high IL-6R expression had poorer survival outcome. In vitro and in vivo studies confirmed that targeting IL-6R with Tocilizumab reduced tamoxifen resistance, providing the basis for potential use for disease management

**Abstract:**

Breast cancer is the most common female cancer. About 70% of breast cancer patients are estrogen receptor α (ERα) positive (ER+) with tamoxifen being the most commonly used anti-endocrine therapy. However, up to 50% of patients who receive tamoxifen suffer recurrence. We previously identified BQ323636.1 (BQ), a novel splice variant of NCOR2, can robustly predict tamoxifen resistance in ER+ primary breast cancer. Here we show that BQ can enhance IL-6/STAT3 signalling. We demonstrated that through interfering with NCOR2 suppressive activity, BQ favours the binding of ER to IL-6 promoter and the binding of NF-ĸB to IL-6 receptor (IL-6R) promoter, leading to the up-regulation of both IL-6 and IL-6R and thus the activation of STAT3. Knockdown of IL-6R could compromise tamoxifen resistance mediated by BQ. Furthermore, Tocilizumab (TCZ), an antibody that binds to IL-6R, could effectively reverse tamoxifen resistance both in vitro and in vivo. Analysis of clinical breast cancer samples confirmed that IL-6R expression was significantly associated with BQ expression and tamoxifen resistance in primary breast cancer, with high IL-6R expression correlating with poorer survival. Multivariate Cox-regression analysis confirmed that high IL-6R expression remained significantly associated with poor overall as well as disease-specific survival in ER+ breast cancer.

## 1. Introduction

Breast cancer is the most common cancer in women worldwide, with nearly 1.7 million new cases diagnosed in 2012. There are five distinct molecular subtypes of breast cancer: luminal A and B, human epidermal growth factor receptor 2 (HER2)-enriched, basal-like, and claudin-low, with the last two being subcategories of triple-negative breast cancer [1]. Each subtype can be targeted differently with systemic therapy. The available systemic therapies include endocrine therapy, targeted biologic treatment, and chemotherapy. Luminal A and B breast cancers are ER-positive subtypes of breast cancer in which estrogen regulates and mediates cell growth. About 70% of breast cancer patients are ER positive (ER+). The estrogen receptor-α (ERα) is a major driver of tumor growth in ER+ breast cancer. ERα cooperates with several other transcription factors to control gene expression and ultimately, tumour growth [2]. ERα-pathway-targeted treatments (e.g., aromatase inhibitors, tamoxifen, and fulvestrant) are the standard of care for patients with this disease [3,4]. However, up to 50% of patients who receive tamoxifen suffer recurrence. Recurrence may be due to the development of de novo or acquired resistance to tamoxifen [5]. Lack of ERα expression is the dominant mechanism of de novo resistance to tamoxifen [6]. Epigenetic changes in the ERα gene may contribute to the development of tamoxifen resistance [5]. Alteration in signalling cascades are essential for development of acquired resistance to tamoxifen. One such mechanism involves cross-talk between ERα and growth factor-mediated signalling pathways [7]. Furthermore, alternative Splicing (AS), which is a predominant mechanism for generating distinct mRNA isoforms from a single gene, also plays an important role in cancer development and treatment [8].

Interleukin-6 (IL-6) is a pleiotropic cytokine that plays a central role in both normal human physiology and disease [9]. IL-6 binds to the receptor complex consisting of IL-6 binding type I transmembrane glycoprotein termed IL-6 receptor-α (IL-6R, CD126) and the type I transmembrane signal transducer protein gp130 (CD130) to activate downstream Janus kinases (JAKs), which subsequently activate Signal transducer and activator of transcription 3 (STAT3) through the phosphorylation of tyrosine 705 [10]. IL-6/STAT3 signalling has been shown to play an important role in tumor progression in many solid tumor types by inducing epithelial-to-mesenchymal transition and angiogenesis [11,12]. IL-6 has been implicated as a driver in pre-clinical models of ER+ breast cancer, and high IL-6 serum and tumor levels have been associated with aggressiveness and poor outcome in patients [13,14,15]. Therefore, the IL-6/STAT3 pathway is a pharmacological target for treating human diseases, cancer included. Multiple approaches have been employed to target this pathway pre-clinically as well as clinically, including targeting of upstream JAKs (e.g., Ruxolitinib), direct targeting of STAT3 phosphorylation and activation (e.g., OPB compounds), and downregulating STAT3 expression (e.g., AZD9150) [16]. Tocilizumab (TCZ; Actemra^®^), which is a humanized monoclonal antibody that binds and neutralizes IL-6R, resulting in the inhibition of IL-6-mediated signalling, has been used clinically to treat rheumatoid arthritis [17].

Expression of IL-6 is governed by Nuclear factor-κB (NF-κB) [18], which is a family of inducible transcription factors, which regulates a large array of genes involved in different processes of the immune and inflammatory responses [19]. This family is composed of five structurally related members, including NF-κB1 (p50), NF-κB2 (p52), RelA (p65), RelB, and c-Rel, which mediates transcription of target genes by binding to a specific DNA element, κB enhancer, as various hetero- or homo-dimers [20]. NF-κB is commonly specifically referred to a p50–p65 heterodimer, which is one of the most avidly forming dimers and is the major NF-κB complex in most cells [20]. NCOR2 (SMRT) is a key co-repressor protein that functions to repress transcription of various transcription factors [21]. Studies have demonstrated that NCOR2 exhibited an interaction with p50 and suppressed the DNA transcriptional activity of p65 [22,23].

We previously identified a novel splice variant of NCOR2 named BQ323636.1 (BQ) and found it to be associated with tamoxifen resistance [24]. Our group examined the expression of BQ by immunohistochemistry in 2095 cases of primary breast cancer in tissue microarray (TMA) from Hong Kong and the United Kingdom. Of the 1271 cases assessed, those with nuclear BQ overexpression showed statistically significant association with poorer survival outcome and tamoxifen resistance, demonstrating that nuclear BQ expression in primary breast cancer could be a robust biomarker for predicting tamoxifen resistance in ER+ breast cancer [25]. Mechanistically, BQ competed with NCOR2 to functionally interfere with the formation of NCOR2 gene co-repressor complex; thus, BQ overexpression compromised the function of NCOR2 in repressing gene expression [25].

In the current study, we demonstrate that BQ overexpression could enhance the expression of IL-6 and IL-6R, which in turn activates the IL-6/STAT3 pathway to mediate tamoxifen resistance in breast cancer. Through in vivo study, we confirmed the clinical significance of IL-6R in breast cancer. By targeting IL-6R, tamoxifen resistance could be reversed both in vitro and in vivo. Our study thus elucidated a potential method for managing tamoxifen resistance in breast cancer.

## 2. Materials and Methods

### 2.1. Cell Culture and Stable Cell Lines Establishment

Human breast cancer cell lines MCF-7 and ZR-75, both which are ER+ and tamoxifen sensitive, were purchased from ATTC and reauthenticated by short tandem repeat profiling [24]. LCC2, a tamoxifen-resistant cell line derived from MCF-7, and AK-47, a tamoxifen resistant cell line derived from ZR-75 cells, were kindly provided by Dr. Robert Clarke (Georgetown University Medical School, Washington, DC, USA) [26]. MCF-7 and LCC2 cells were cultured and maintained in Dulbecco’s Modified Eagle Medium (DMEM, Gibco, Gaithersburg, MD, USA) supplemented with 10% fetal bovine serum (Gibco, Gaithersburg, MD, USA) and 1% penicillin/streptomycin (Gibco, Gaithersburg, MD, USA). ZR-75 and AK-47 cells were grown in Improved Minimum Essential Medium (IMEM, Gibco, Gaithersburg, MD, USA) with the addition of 10% FBS and 1% P/S. MCF-10A was purchased from ATCC and cultured in Mammary Epithelial Cell Growth Medium (Lonza, Morristown, NJ, USA) supplemented by Bovine Pituitary Extract (Lonza), Human Epidermal Growth Factor (Lonza, Morristown, NJ, USA), Hydrocortisone, and 100 ng/mL cholera toxin.

MCF-7 and ZR-75 were transfected with pcDNA3.1 or pcDNA3.1-BQ to establish control cell lines and BQ overexpressing cell lines respectively. The stable cell lines were selected and maintained in the culture medium with 0.5 mg/mL of G418 (Gibco, Gaithersburg, MD, USA). IL-6R Human shRNA Plasmid Kit (TL312161; Origene, Rockville, MD, USA) was employed. MCF-7, ZR-75, LCC2, and AK-47 were transfected with plasmid, which expressed non-targeting shRNA (shCtrl), shRNA1 targeting IL-6R (shIL-6R.1; TL312161A), and shRNA2 targeting IL-6R (shIL-6R.2; TL312161B). The stable cell lines were selected and maintained in the culture medium with 1 µg/mL of puromycin (Gibco, Gaithersburg, MD, USA). IL-6 CRISPR/Cas9 KO Plasmid (sc-400390; Santa Cruz Biotechnology, Dallas, TX, USA) and IL-6 HDR Plasmid (sc-400390-HDR; Santa Cruz Biotechnology) were employed to generate IL-6 knockout (KO) cell lines. MCF-7-BQ and ZR-75-BQ were co-transfected with IL-6 CRISPR/Cas9 KO Plasmid and IL-6 HDR Plasmid. The stable cell lines were selected and maintained in the culture medium with 1 µg/mL of puromycin (Gibco, Gaithersburg, MD, USA) and 0.5 mg/mL of G418 (Gibco, Gaithersburg, MD, USA). All the cell lines were cultured in a tissue culture incubator with 5% CO_2_ at 37 °C. All cell lines were confirmed free of mycoplasma (service provided by core facilities, The University of Hong Kong).

### 2.2. Chemicals

Human IL-6 Recombinant Protein (RP-8619; Invitrogen, Waltham, MA, USA) and bovine serum albumin (BSA; J10857-36; Thermo Scientific, Waltham, MA, USA) were dissolved in double-distilled water. STAT3 Inhibitor VI S3I-201 (sc-204304; Santa Cruz Biotechnology, Dallas, TX, USA) was dissolved in DMSO. Tocilizumab (TCZ; Actemra^®^; Roche, Switzerland) was used and diluted in saline. β-Estradiol (E_2_; E8875; Sigma, St. Louis, MO, USA) was dissolved in 100% ethanol. NF-κB inhibitor p-XSC (ab142600; Abcam, Cambridge, UK) was dissolved in DMSO. Tamoxifen (TAM/4-OHT; H6278; Sigma, St. Louis, MO, USA) was dissolved into 100% ethanol (EtOH).

### 2.3. Cell Viability Assay

MTT assay (3-(4,5-Dimethylthiazol-2-yl)-2,5-Diphenyltetrazolium Bromide; M6494; Invitrogen) was performed according to the manufacturer’s instructions. Absorbance was recorded by Tecan Infinite F200 plate-reader. Clonogenic assay was performed by staining the cells with 0.01% of crystal violet (C0775; Sigma, St. Louis, MO, USA). Any colony with more than 50 cells was regarded as one colony.

### 2.4. ELISA Assay

IL-6 Human ELISA Kit (EH2IL6; Invitrogen, Waltham, MA, USA) was used according to the instruction manual. Absorbance at 450 nm was recorded by Tecan infinite F200 platereader.

### 2.5. Gene Silencing, Plasmids, qPCR, Gene Expression Analysis and Promoter Analysis

siBQ.1 (5′-CUU CUC CAG GUU CUC UGC AUG-3′) and siBQ.2 (5′-CUC CAG GUU CUC UGC AUG CGC-3′) were purchased (Sigma). Negative control siRNA (siCtrl; 4390843; ThermoFisher, Waltham, MA, USA) and IL-6 siRNA (siIL-6; s7311; ThermoFisher, Waltham, MA, USA) were employed. Ten pmol of the siRNA was used. Oligofectamine (12252011; Invitrogen, Waltham, MA, USA) was used for the delivery of the siRNA. TRIzol reagent (15596026; Invitrogen, Waltham, MA, USA) was used for total RNA extraction following manufacturer’s protocol. Up to 0.5 µg total RNA were reverse transcribed into cDNA by SuperScript III reverse transcriptase (18080093; Invitrogen, St. Louis, MO, USA) following the manufacturer’s protocol. Overexpression of NCOR2 was mediated by transfection of pCMV6_NCOR2 (RC212113; Origene, MD, USA) using Lipofectamine 2000 (11668-019; Invitrogen, Waltham, MA, USA). Applied Biosystems 7900HT was used. qPCR was employed to determine the relative expression of genes. RT^2^ Profiler™ PCR Array Human Cancer PathwayFinder™ (PAHS-033Z; Qiagen, Germany) was used. Power SYBR Green Master Mix (A25742; Applied Biosystems, Foster City, CA, USA) was used. ΔΔCT method was used to determine relative gene expression level. Actin was used as the internal control. Untreated control was used as the reference. Pathway enrichment analysis was preformed using KEGG database. The following primers (5′→3′) were used: actin-F (ATC GTG CGT GAC ATT AAG GAG AAG) and actin-R (AGG AAG GAA GGC TGG AAG AGT G); IL-6-F (ATA ACC ACC CCT GAC CCA AC) and IL-6-R (GAA CTC CTT AAA GCT GCG CA). IL-6R-F (AGA CAG GTG CGA AAG GAT GA) and IL-6R-R (TCT ACA GAC AAG CCC AGC AA) BQ-F (AAG GTG GAG CGC ATC GAG AAC) and BQ-R (GCA TCT GCT TCT CCA GGT TCT CTG); NCOR2-F (ACG AGG TGT CAG AGA TCA TCG A) and NCOR2-R (TGA TGA ACT TGA TGC GCT GCT). DNA sequence between +100 to −1000 was retrieved using The Eukaryotic Promoter Database on 10 June 2020 (https://epd.epfl.ch//index.php) [27]. ER binding site and NF-ĸB1 binding site were examined in IL-6 and IL-6R promoters respectively. The location of transcription initiator was determined within the promoter region.

### 2.6. Chromatin Immunoprecipitation (ChIP)

Cell lysates were fixed for 10 min with 0.5% paraformaldehyde and suspended in 200 μL of lysis buffer (20 mM Tris-Cl, pH 7.4, 150 mM NaCl, 5 mM MgCl_2_, and 0.5% Nonidet P-40). The suspended cells were lysed by sonication to yield DNA fragments of 0.1–5 kb using Diagenode BioruptorPico UltraSonication System (Centre for PanorOmic Sciences, HKU). Protein A agarose beads (20333; Thermo Scientific, Waltham, MA, USA) were precleared for 1 h at 4 °C in the presence of 100 μg/mL salmon sperm DNA (AM9680; Invitrogen, Waltham, MA, USA). Antigens were immunoprecipitated overnight at 4 °C with the following antibodies: anti-p50 (1:200; 13586; Cell signaling Technology, Danvers, MA, USA), Anti-ERα (1:200; 8644; Cell signaling Technology, Danvers, MA, USA). Ten percent of cell lysate was saved as input control. The agarose beads were washed in lysis buffer three times each for 10 min before elution. Cross-links were reversed by heating for 6 h at 65 °C. The DNA fragments were then eluted and purified by QIAquick PCR purification kit (28104; Qiagen, Germany). The purified DNA fragments were amplified by PCR, using the following primers (5′→3′): IL-6-ERE1-F (CAT GCC AAA GTG CTG AGT CA) and IL-6-ERE1-R (AGT GCA GCT TAG GTC GTC AT); IL-6-ERE2-F (CCC TCA CCC TCC AAC AAA GA) and IL-6-ERE2-R (GAG CTT CTC TTT CGT TCC CG); IL6R-P50-RE-F (AGT CCA AAC CGT TTC CTT GC); and IL6R-P50-RE-R (GGC GAT GTT CCT CTT ACC CT).

### 2.7. Tissue Microarray

One-hundred-and-forty-one cases of breast cancer diagnosed between the years 1993–2003 with clinical follow up data were retrieved from the records of the Department of Pathology, Queen Mary Hospital of Hong Kong, with approval (UW 08-147) by the Institutional Review Board of The University of Hong Kong. The status of ER, PR, and HER2 were obtained from pathological reports from Queen Mary Hospital. ER, PR, and HER2 expression in the primary tumors were examined by IHC. Those cases with equivocal HER2 expression were further examined by FISH. Histological sections of these cases were reviewed by the pathologist; the representative paraffin tumor blocks were chosen as donor block for each case. Selected areas were marked for construction of tissue microarray (TMA) blocks. A total of 137 cases could be assessed (Table 1). Of these, there were 73 ER+ cases (Table 2) that had been given adjuvant tamoxifen treatment with available follow-up clinical data. Tamoxifen resistance is defined as patients who had been treated with tamoxifen in the adjuvant setting but subsequently developed disease relapse or distant metastases.

### 2.8. Immunohistochemistry

The IHC was performed as previously described [25]. Anti-BQ antibody (D-12; Versitech LTD, Telegraph Bay, Hong Kong) and anti-IL-6R antibody (A101411; Antibodies.com, Cambridge, UK) were diluted at 1:50 and 1:30 respectively. Aperio ImageScope^®^ system (Leica Biosystems Aperio, Nußloch, Germany) was used to assess BQ323636.1 and IL-6R expression and scored by two independent individuals. For BQ staining, the intensities and percentages of nuclear staining were assessed using H-scoring system. H-score = (1 × % of cells stained at intensity category 1) + (2 × % of cells stained at intensity category 2) + (3 × % of cells stained at intensity category 3). As IL-6R is localized to the cytoplasm, the intensities and percentages of cytoplasmic staining were assessed as follows: Intensity was scored as 0 = none, 1 = weak, 2 = moderate, and 3 = strong. The percentage of cells stained was scored as 1 = less than 25%, 2 = >25% to 50%, 3 = >50% to 75%, and 4 = >75%. The cytoplasmic score was calculated as the product of the percentage score and the intensity score. The median value of H-score and cytoplasmic score were used to dichotomize the cut-off between low and high expression. These were 110 and 6 for BQ and IL-6R respectively.

### 2.9. Luciferase Reporter Assay

STAT3 Reporter Kit (79730; BPS Bioscience, San Diego, CA, USA) and NF-κB Reporter Kit (60614; BPS Bioscience, San Diego, CA, USA) were employed and Lipofectamine 2000 (11668-019; Invitrogen, Waltham, MA, USA) was used for the transfection. Dual-Luciferase^®^ Reporter Assay System was employed (E1910; Promega, Madison, WI, USA). Chemoluminance signal was recorded by Tecan infinite F200 plate reader.

### 2.10. Preparation of Conditional Medium

MCF-7, MCF-7-BQ, MCF-7-BQ siCtrl, MCF-7-BQ siIL6, ZR-75, ZR-75-BQ, ZR-75-BQ siCtrl, ZR-75-BQ siIL6, LCC2, and AK-47 cell lines were cultured in the medium (phenol-red free) supplement with 10% of charcoal-stripped FBS (Gibco, Gaithersburg, MD, USA) and 1% penicillin/streptomycin (Gibco, Gaithersburg, ML, USA). After 72 h, the medium was collected and centrifuged for 10 min at 4 °C with 4000 rpm by Heraeus multifuge ×3 FR centrifuge (rotor: 75003180; Thermo Scientific, Waltham, MA, USA) to remove cell debris. The conditional medium (CM) was stored at −20 °C in aliquots.

### 2.11. Tamoxifen Response Assay

The cells were treated with 5 µM of tamoxifen (4-OHT; H6278; Sigma, St. Louis, MO, USA). 4-OHT was dissolved into 100% ethanol (EtOH). MTT and clonogenic assays were employed to determine the proportion of viable cells.

### 2.12. Western Blot and Co-Immunoprecipitation

Cell pellets were lysed using cell lysis buffer prepared by mixing 1× Cell Lysis Buffer (9803; Cell Signaling Technology, Danvers, MA, USA), 10% glycerol, cOmplete Mini, EDTA-free protease inhibitor cocktail (11873580001; Roche, Switzerland) and PhosSTOP EASYPack tablets (4906845001; Roche, Switzerland). Cell pellets were lysed in 200 µL of IP buffer (20 mM Tris-Cl, pH7.4, 150 mM NaCl, 5 mM MgCl2, 0.5% NP-40, 10% glycerol). Twenty microliters was stored as input. The remaining cell lysates were divided into two portions, 90 µL each. Anti-ERα (1:200; 8644; Cell signalling Technology, Danvers, MA, USA) and anti-NCOR2 (1:200; ab24551l; Abcam, Cambridge, UK) were added and incubated at 4 °C overnight with rotation. Anti-mouse/rabbit IgG was used as negative control. The immunoprecipitant was incubated with 50 µL of Protein A/G agarose beads (20421; Thermo Scientific, Waltham, MA, USA) at room temperature for 2 h with rotation. The beads were washed three time with 1 mL of IP buffer, 10 min each. Protein A/G HRP (1:8000; 32490; Pierce, Rockford, IL, USA) was used as secondary antibody. The protein concentration of the cell lysates was determined by DC protein assay (1620177; BioRad, Hercules, CA, USA). SDS-polyacrylamide gels were made following the protocol for Western blot analysis. Twenty micrograms of total proteins were loaded in each of the wells of the gel. The proteins were transferred onto PVDF membrane (1620177; BioRad, Hercules, CA, USA). The signal was captured Amersham Imager 680 (GE Healthcare). Uncropped blots were shown in Appendix A. The follow antibodies were used: anti-HIS tag (1:4000; #2366; Cell Signaling Technology); anti-BQ (1:500; D-12; Versitech Ltd., Telegraph Bay, Hong Kong); anti-STAT3 (1:1000; 9139; Cell Signaling Technology, Danvers, MA, USA); anti-pSTAT3 (1:1000; 9131, Cell Signaling Technology, Danvers, MA, USA); anti-IL-6R (1:2000; ab128008; Abcam, UK); anti-NCOR2 (1:1000; ab24551l; Abcam, UK); anti-p50 (1:2000; 13586; Cell Signaling Technology, Danvers, MA, USA), anti-p65 (1:1000; 8242; Cell Signaling Technology, Danvers, MA, USA); anti-tubulin (1:10,000; 2146; Cell Signaling Technology, Danvers, MA, USA); and anti-actin (1:10,000; sc-47778; Santa Cruz Biotechnology, Dallas, TX, USA). The following reagents were used to generate signal: anti-mouse HRP (1:5000; P0447; Dako, Denmark), anti-rabbit HRP (1:5000; P0260; Dako, Denmark).

### 2.13. Xenograft

Female nude mice, aged 5 to 6 weeks, were used for this study. 1 × 10^7^ cells were mixed with Matrigel (356234; BD Bioscience, Franklin Lakes, NJ, USA) at a ratio of 1:1 and the 100 µL cell mixture injected into the abdominal mammary fat pad of mice. When the tumors were palpable, mice were randomized into treatment and control groups where the treatment group received 0.5 mg of 4-OHT tamoxifen (H6278; Sigma, St. Louis, MO, USA) dissolved in ethanol and diluted in peanut oil (P2144; Sigma, St. Louis, MO, USA), and TCZ (1 mg/Kg and 2 mg/Kg) diluted in saline, given by subcutaneous injection. The mice were treated twice a week for 8 weeks.

### 2.14. Statistical Analysis

All numerical data were processed in Excel (Microsoft), Prism5 (GraphPad) or SPSS25 (IBM). All data were expressed as mean ± SD from at least three independent experiments. Mann-Whitney U test or Students’ *t* test were performed to compare the variables of the two sample groups. All tests were two-sided unless otherwise specified. The data were dichotomized into two groups including high or low expression using median expression level as cut-off. The correlations were analyzed by Chi-square tests. The expression levels of BQ, IL-6R were compared between different groups using Mann–Whitney U Rank test. Survival analyses were done by Kaplan–Meier estimates followed by Log-rank test and Cox regression model. *p* value of less than 0.05 was considered statistically significant.

## 3. Results

### 3.1. Overexpression of BQ Enhanced IL-6/STAT3 Signalling Pathway

Our previous studies had confirmed that BQ overexpression resulted in tamoxifen resistance in breast cancer cells both in vitro and in vivo [25]. In addition, we confirmed that tamoxifen-resistant cells have a higher ratio of BQ to NCOR2 (Appendix A). To elucidate the molecular mechanisms by which BQ could contribute to the resistance, we employed Cancer PathFinder PCR array to determine the effect of BQ on the expression of 84 cancer-related genes. The comparison was made between two pairs of cell lines, MCF-7-BQ versus MCF-7 and ZR-75-BQ versus ZR-75. From the expression profiles (Figure 1A and Appendix A), we observe that BQ overexpression enhanced the expression of a certain subset of genes common for both MCF-7 and ZR-75. KEGG pathway enrichment analysis identified seven common pathways significantly enriched (*p* < 0.05) for both MCF-7 and ZR-75 (Figure 1B and Appendix A), with the IL-6/STAT3 signalling pathway one of the significantly enriched pathways involved in both cell lines. qPCR independently validated that overexpression of BQ could enhance mRNA IL-6 expression in both of the cell lines (Figure 1C), whilst ELISA assay confirmed BQ overexpression could enhance the production of IL-6 (Figure 1D). IL-6 is known to enhance the activity of STAT3 through triggering phosphorylation on STAT3 [11]. We confirmed that BQ overexpression could indeed enhance the level of phosphorylated STAT3 through Western blot (Figure 1E) and could enhance the transcriptional activity of STAT3 through luciferase reporter assay (Figure 1F). On the other hand, we designed two independent siRNA to reduce the expression of BQ in LCC2 cells (Figure 1G), which had no effect on NCOR2 mRNA expression (Appendix A). Through STAT3 luciferase reporter assay, we found that down-regulation of BQ could suppress STAT3 activity in LCC2 (Figure 1H). Moreover, addition of IL-6 in LCC2 treated with siBQ.1 and siBQ.2 could rescue the activity of STAT3 (Appendix A). Based on these findings, we confirmed that overexpression of BQ could modulate IL-6/STAT3 signalling pathway.

### 3.2. Activation of IL-6/STAT3 Pathway Could Induce Tamoxifen Resistance

We next confirmed that tamoxifen resistance could be induced by the activation of IL-6/STAT3 signalling. First, we collected conditional medium (CM) from MCF-7-BQ and ZR-75-BQ cells and used the CM to treat MCF-7 and ZR-75 together with tamoxifen. After 96-h, cell viability was determined by MTT assay. The results showed that the CM from BQ-overexpressing cells could induce the development of tamoxifen resistance (Figure 2A). Similarly, we collected CM from LCC2 and AK-47, which expressed high endogenous BQ. As expected, the CM could induce tamoxifen resistance in MCF-7 and ZR-75 (Figure 2B). Next, we employed 20 pmol of siRNA against IL-6 or non-targeting siRNAs to treat MCF-7-BQ, ZR-75-BQ, LCC2, and AK-47 (Appendix A). The culture medium from these siRNA-treated cells were collected and used to cultivate with MCF-7 and ZR-75 cells. ELISA assay confirmed that the IL-6 siRNA could reduce the production of IL-6 significantly (Figure 2C). The results from tamoxifen response assay showed that down-regulation of IL-6 could compromise the effect on the induction of tamoxifen resistance (Figure 2D). Moreover, knockout of IL-6 (Appendix A) in MCF-7-BQ and ZR-75-BQ could reverse tamoxifen resistance as revealed by MTT (Figure 2E). Next, we employed recombinant IL-6 to test whether the treatment of IL-6 directly could induce tamoxifen resistance. We determined the optimal dosage of IL-6 by examining its effect on cell viability and STAT3 activity. The results showed that 10 ng/mL of IL-6 should be the optimal dosage as it would not affect cell viability (Appendix A) and it resulted in significantly increased STAT3 transcriptional activity (Appendix A) as revealed by MTT and luciferase reporter assays respectively. The treatment of 10 ng/mL of IL-6 could induce tamoxifen resistance in MCF-7 and ZR-75 (Figure 2F,G). Subsequently, we employed STAT3 inhibitor S3I-201 to suppress STAT3 transcriptional activity. One micrometre of S3I-201 was the highest non-lethal concentration (Appendix A). As expected, the additional of S3I-201 could abolish the effect of IL-6-induced tamoxifen resistance (Figure 2H).

### 3.3. Overexpression BQ Could Enhance the Transcription of IL-6 Mediated by Estrogen Receptor α (ER)

From our promoter analysis, we found seven potential ERα binding sites (estrogen response element; ERE) in the promoter region of IL-6, with two of them (−120 and +2) close to the transcription initiation site (Appendix A). We therefore hypothesized that ERα in breast cancer cells could mediate the expression of IL-6. We treated MCF-7 and ZR-75 cells with 1 nM of Estrogen (E_2_). Results from qPCR showed that while E_2_ treatment suppressed IL-6 expression, its effect was opposite when BQ was overexpressed (Figure 3A). ChIP assay was performed on MCF-7 and ZR-75 to determine whether 1 nM of E_2_ would affect the binding of ERα to region 1 and region 2 of the IL-6 promoter. The results showed that treatment of E_2_ could only enhance the binding of ERα to region 1 (Figure 3B) but not to region 2 (Figure 3C). ChIP assay also showed that overexpression of BQ could further enhance the interaction between ERα and region 1 in the presence of E_2_ (Figure 3D). Furthermore, the expression of IL-6 in the BQ overexpressing cells could be suppressed by gradually increasing the expression of NCOR2 (Figure 3E). Our results suggested that BQ overexpression could enhance the expression of IL-6 through the activity of ERα. Together with our previous study [25], we therefore identified a novel molecular pathway to modulate IL-6 expression in ER+ breast cancer (Figure 3F).

### 3.4. Overexpression BQ Could Enhance the Transcription of IL-6R Mediated by NF-kB

In order for IL-6/STAT3 signalling to be activated, IL-6 receptor (IL-6R) must be present. We hypothesized that BQ overexpression might enhance the expression of IL-6R in breast cancer and was able to confirm that BQ overexpression could indeed enhance the expression of IL-6R in breast cancer cells at both mRNA (Figure 4A) and protein (Figure 4B) levels. To address the molecular mechanism through which BQ could enhance the expression of IL-6R, we analysed the promoter region of IL-6R and found the p50 (NF-kB1) binding site to be close to the transcription initiator (Appendix A). ChIP assay confirmed that p50 could indeed interact with the p50 binding element in the IL-6R promoter region (Figure 4C). Since p50 is known to interact with NCOR2 [23], this led us to postulate that BQ, a splice variant of NCOR2, might modulate NF-κB activity through competition with NCOR2. We first confirmed that NCOR2 could form a protein complex with p50–p65 in breast cancer cells as shown by co-immunoprecipitation (Figure 4D) and next demonstrated that BQ could compromise the interaction between NCOR2 and p50–p65 (Figure 4E). These results are supportive that BQ can interfere with the repressive effect of NCOR2 on NF-κB and suggest that BQ overexpression might modulate IL-6R expression by compromising the repressive activity of NCOR2 [23]. To further validate the importance of NF-κB for IL-6R expression, we examined the effect of NF-κB inhibitor pXSC, which can covalently modify p50 to abolish its DNA-binding ability. Having first determined that 200 nM of the inhibitor will not affect cell viability of both MCF-7 and ZR-75 (Appendix A), we demonstrated by qPCR that treatment of the NF-κB inhibitor could indeed suppress IL-6R expression in the BQ-overexpressing cells (Figure 4F). These findings confirm that BQ requires NF-κB to modulate the expression of IL-6R in breast cancer (Figure 4G).

### 3.5. Targeting IL-6R Could Reduce Tamoxifen Resistance

We next determined the effect of IL-6R knockdown (Appendix A) on tamoxifen resistance in BQ-overexpressing cells. The results from tamoxifen response assay showed that whilst knockdown of IL-6R itself did not alter tamoxifen response of tamoxifen-sensitive MCF-7 and ZR-75 cells, IL-6R knockdown could reverse the tamoxifen resistance of BQ-overexpressing cells to make them sensitive to tamoxifen (Figure 5A). As expected, IL-6R knockdown could compromise the effect of IL-6 on STAT3 activity (Appendix A). These results suggest that through targeting IL-6R, it may be possible to inhibit IL-6/STAT3 signalling and thus reduce tamoxifen resistance in breast cancer. To address this issue, we employed TCZ, a monoclonal antibody, to bind to IL-6R. We established TCZ of 250 ng/mL as the maximum non-lethal concentration (Appendix A) and found that the addition of TCZ could reverse tamoxifen resistance in MCF-7-BQ and ZR-75-BQ cells as revealed by MTT (Figure 5B) and clonogenic assay (Figure 5C,D). In addition, we examined whether TCZ could reverse tamoxifen resistance in an animal model. We first generated xenograft models established from ZR-75-BQ and treated the mice with 1 mg/Kg and 2 mg/Kg of TCZ together with tamoxifen for 2 months. The results showed that the addition of TCZ could make the tumours become sensitive to tamoxifen in a dose-dependent manner (Appendix A). Next, we determined the effect of TCZ on the xenograft models established from tamoxifen-resistant cell line LCC2, which had a high expression of endogenous BQ. We treated the mice with 2 mg/Kg of TCZ for 2 months. The results showed that the addition of TCZ could reverse tamoxifen resistance (Figure 5E,F). Such an effect was IL-6R-dependent, as knockdown of IL-6R could abolish the effect of TCZ on tamoxifen response in vivo. Based on our study, we confirmed that targeting IL-6R should be a possible strategy to reverse tamoxifen resistance in breast cancer.

### 3.6. Clinical Significance of BQ and IL-6R in Breast Cancer

To provide further in vivo evidence to support our findings, we correlated the expression of BQ and IL-6R in clinical breast cancer samples. Immunohistochemistry (IHC) was performed on TMA of primary breast cancer cases to examine the expression of BQ and IL-6R in breast cancer tissues. BQ expression is functional in the nucleus, hence nuclear expression was assessed by H-score, whilst for IL-6R expression, being localized to the cytoplasm, the cytoplasmic score was used. The median values of both scores were used to dichotomize into low and high expression groups (Figure 6A). Correlating BQ nuclear with IL-6R cytoplasmic expressions in ER+ breast cancer, we found statistically significant direct correlation between IL-6R and BQ expression (*p* = 0.027 Mann–Whitney U test), as well as positive correlation by chi-square test (*p* = 0.011) (Figure 6B). Moreover, high expression of IL-6R was significantly correlated with tamoxifen resistance in these ER+ breast cancer cases (*p* = 0.005 Mann–Whitney U test), as well as by chi-square test (*p* = 1.9 × 10^−5^) (Figure 6C). This was also demonstrated by Kaplan–Meier survival analysis showing patients with high expression of IL-6R with poorer outcome for both overall survival (*p* = 0.002; Figure 6D) and disease-specific survival (*p* = 0.005; Figure 6E). By univariate cox regression analysis, high expression of IL-6R was significantly associated with poorer overall survival (*p* = 0.004; RR: 3.716 95% CI: 1.537, 8.984; Table 3) as well as poorer disease-specific survival (*p* = 0.008; RR: 5.664 95% CI: 1.569, 20.441; Table 4). Multivariate cox-regression analyses showed that after adjustment for the other variables, high expression of IL-6R remained significantly associated with poorer overall survival as well as disease-specific survival (*p* = 0.036; RR: 10.967 95% CI: 1.169, 102.878; Table 3); (*p* = 0.009; RR: 5.586 95% CI: 1.534, 20.349; Table 4) respectively. Altogether, our results confirmed high expression of IL-6R in ER+ breast cancer was associated with tamoxifen resistance and poorer survival outcome; and IL-6R could be an independent prognostic factor.

## 4. Discussion

Our study illustrates that overexpression of BQ can modulate tamoxifen resistance by enhancing the expression of IL-6 and IL-6R in ER+ breast cancer. Through molecular studies, we illustrated that BQ through interacting with NCOR2 could interfere with its function. NCOR2 itself is a central component for mediating gene repression [28]. NCOR2 functions as a dimer in the corepressor complex, recruiting other corepressor proteins such as GPS, TBLR1, and HDAC3 to mediate the repression of transcription factor activity to suppress gene expression. Our previous study identified BQ, a novel splice variant of NCOR2, which retains the N-terminus of NCOR [24]. We showed that BQ could interact with NCOR2 and this interaction could disrupt the interaction between NCOR2 with other regulatory factors, thus targeting transcription factors such as ERα [25]. In the current study, we identified two potential ERα binding sites within the promoter region of IL-6 (Appendix A). Previous studies have shown that the addition of estrogen could impair the expression of IL-6/STAT3 signalling pathway through ER and STAT3 modulator PIAS [29,30]. One of these studies found that activation of ER could occupy the NF-κB binding site in the promoter of IL-6, thus blocking the transcription of IL-6.

In our study, we observed that overexpression of BQ reversed the effect of ER on IL-6 expression as demonstrated by the enhanced IL-6 expression shown in Figure 3A. In the presence of BQ overexpression, ER could bind to ERE in the IL-6 promoter and the addition of estrogen could further enhance the proportion of ER binding to this region (Figure 3D). We believe this enhanced binding in turn enhanced the expression of IL-6. One possible reason could be the interaction between BQ and NCOR2. NCOR2 is a repressor of gene expression; and the interaction between NCOR2 and ER is believed to suppress the initiation of transcription [22]. BQ overexpression, in interacting and competing with NCOR2, would diminish the suppressive role of NCOR2 on IL-6 expression in the presence of ER.

Interleukin-6 (IL-6) is a multipotent cytokine that plays an important role in immune responses and human diseases, including different type of cancers [31]. A high concentration of serum IL-6 has been shown to be associated with aggressive tumour types and poor disease-free and overall survival [14]. IL-6 binds to IL-6 receptor (IL-6R) to activate STAT3. Constitutive activation of STAT3 has frequently been observed in a variety of tumours, including breast cancer [32], and such activation can promote proliferation and survival of cancer cells [33]. Recent reports showed that STAT3 activation is associated with drug resistance outcomes, and that blockade of STAT3 pathway can restore the efficacy of chemodrugs. Through activation of the STAT3/SNAIL pathway, cisplatin treatment rendered drug resistance, enhanced the EMT-like phenotype, and increased migration and invasion abilities in tumour cells [34]. In addition, STAT3 activation also mediates resistance to chemotherapeutic agents 5-fluorouracil (5-FU), oxaliplatin, and SN-380 [35], suggesting that STAT3 activation is responsible for such resistance. In breast cancer, STAT3 activation is associated with tamoxifen resistance [36]. Therefore, inhibition of STAT3 signaling will be an attractive strategy to combat drug resistance.

In our study, we confirmed that overexpression of BQ could lead to the activation of IL-6/STAT3 pathway (Figure 1), which conferred resistance to tamoxifen in ER+ breast cancer cells (Figure 2). The activation process required the presence of ERα and NF-ĸB that modulated the expression of IL-6 (Figure 3) and IL-6R (Figure 4) respectively. Instead of direct inhibition of STAT3 via any small inhibitor, we proposed to suppress STAT3 activity by compromising its activation process. We showed that Tocilizumab (TCZ), an FDA-approved anti-IL-6R antibody for the treatment of patients with rheumatoid arthritis, could reverse tamoxifen resistance in vitro and in vivo (Figure 5). We have successfully illustrated that targeting IL-6R could be an alternative way to suppress IL-6/STAT3 pathway in breast cancer and demonstrate the usage of TCZ could reduce tamoxifen resistance in breast cancer.

To assess the usefulness of TCZ in overcoming tamoxifen resistance, we first ascertained the expression of IL-6R in breast cancer tissue. The protein expression of IL-6R was determined by IHC on TMA of primary breast cancer (Figure 6A), and the expression level of IL-6R was shown to be significantly correlated with tamoxifen resistance (Figure 6C). In vitro study demonstrated that BQ could modulate the expression of IL-6R (Figure 4), while in vivo we found a positive correlation between BQ and IL-6R (Figure 6B). Moreover, high expression of IL-6R showed significant association with poorer prognosis as indicated both by Kaplan–Meier (Figure 6D–G) and cox regression (Table 2) survival analyses. These results support the clinical significance of IL-6R in breast cancer. Thus, our study not only confirms that IL-6R could be an independent prognostic factor in ER+ breast cancer, but could also be a possible target to suppress tamoxifen resistance.

## 5. Conclusions

Our study has elucidated the molecular mechanism through which BQ can modulate the expression of IL-6 and IL-6R and thus the activation of STAT3 in breast cancer. This mechanism could contribute to the development of tamoxifen resistance in breast cancer. We have shown that targeting the IL-6/STAT3 signalling pathway by TCZ could successfully reverse tamoxifen resistance in vitro and in vivo. Our results highlight the significance of IL-6R in ER+ breast cancer and provides the basis for the development of a novel strategy for reversing tamoxifen resistance in breast cancer. Since TCZ has approval for clinical use in various immunological inflammatory conditions, it might thus be more easily considered for clinical trial in the management of breast cancer patients.

## Figures and Tables

**Figure 1 cancers-13-01511-f001:**
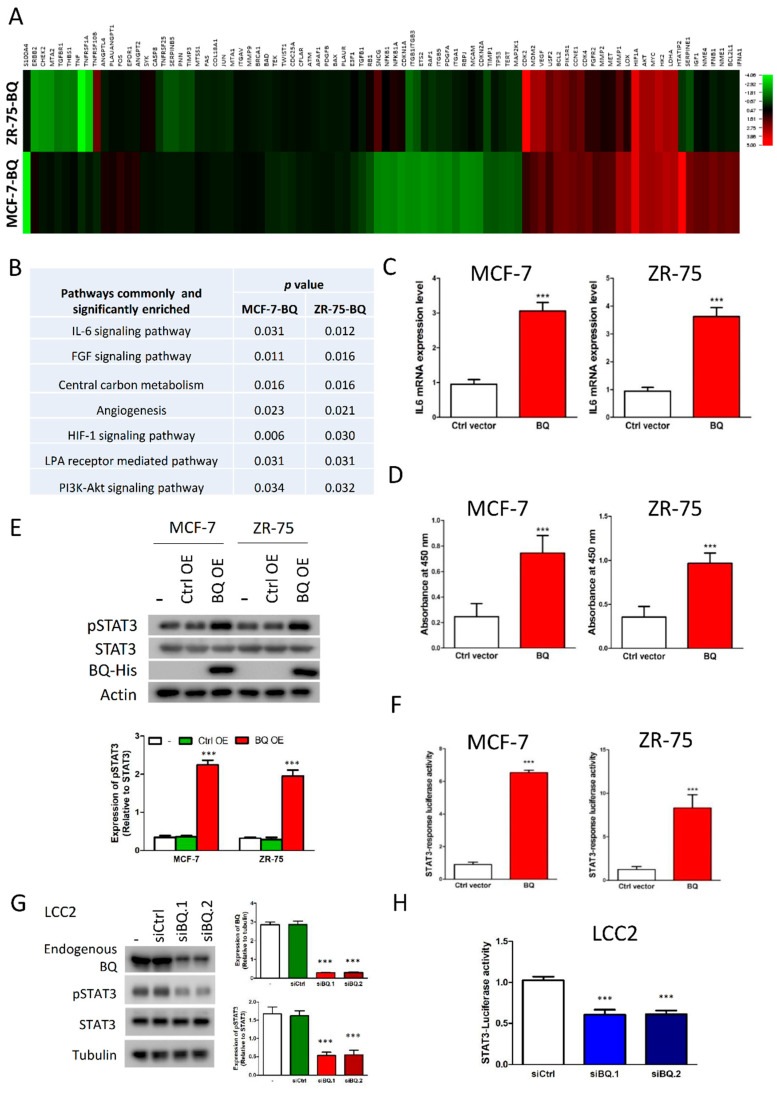
Overexpression of BQ could enhance IL-6/STAT3 signalling pathway in breast cancer cells. (**A**) The Heatmap showed the expression profiles of 84 cancer-related genes. The comparison was made between BQ overexpressing cells and control cells. Gene expression was compared between BQ overexpressing cells (MCF-7-BQ and ZR-75-BQ) and control cells (MCF-7 and ZR-75). Relative expression was determined and the value of expression data was subjected to log_2_ transformation for plotting the Heatmap. (**B**) Signaling pathways commonly and significantly enriched (*p* < 0.05) in the two BQ overexpressing breast cancer cell lines. KEGG pathway enrichment analysis was performed to identify the enriched pathways. (**C**) Overexpression of BQ could enhance mRNA expression of IL-6. MCF-7 and ZR-75 were transfected with 0.5 µg of *pcDNA3.1_BQ* or *pcDNA3.1*. qPCR was performed 72 h post transfection. Untransfected cells were used as reference. Actin was used as internal control. (**D**) Overexpression of BQ could enhance the production of IL-6. ELISA was performed on the culture medium to detect the level of IL-6. (**E**) Overexpression of BQ could enhance the expression level of phosphorylated STAT3 (pSTAT3). Western blot was performed to detect the indicated proteins. (**F**) Overexpression of BQ could enhance the transcriptional activity of STAT3. Luciferase reporter assay was performed 72 h post transfection to determine STAT3 activity. (**G**) Knockdown efficiency of siRNA against BQ. LCC2 cells were treated with two independent siRNAs. 20 pmol of the siRNA was used. Western blot was performed 72 h post transfection. Tubulin was used as loading control. (**H**) Knockdown of BQ could suppress STAT3 activity in LCC2 cells. Luciferase reporter assay was performed 72 h post-transfection to determine STAT3 activity. Results were shown as mean ± SD from at least three independent experiments. Student’s *t* test was used to determine the statistical significance between two groups. *** represents *p* < 0.001.

**Figure 2 cancers-13-01511-f002:**
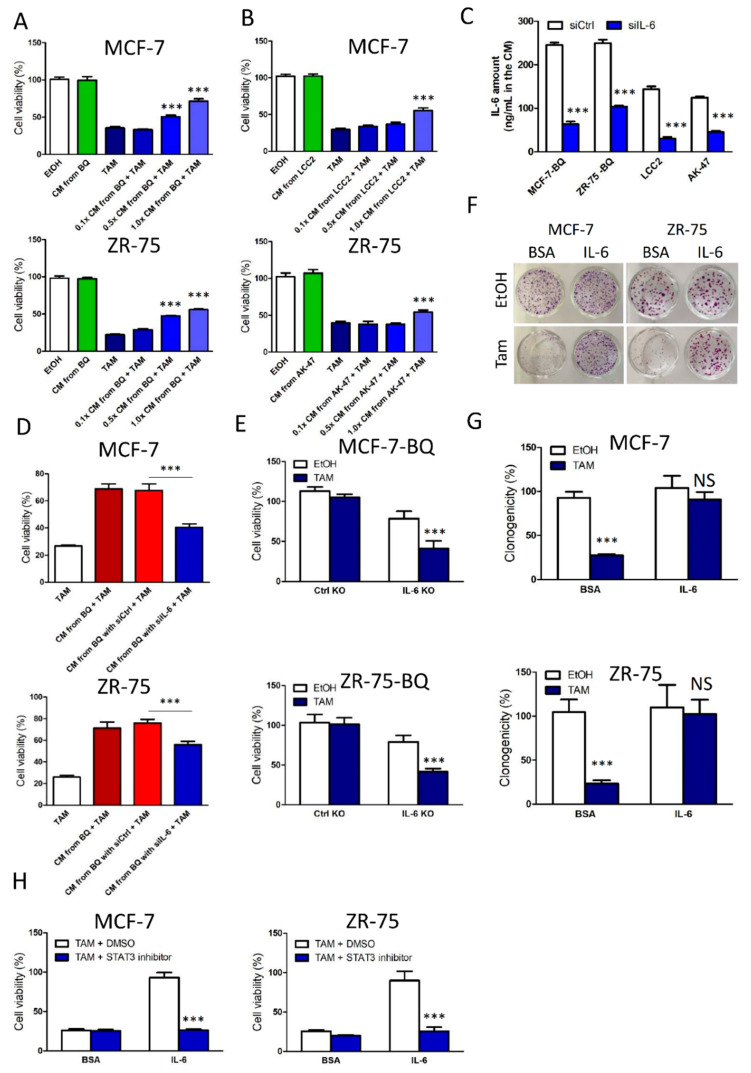
IL-6/STAT3 signalling pathway could modulate tamoxifen response in ER+ breast cancer cells. (**A**) Conditional medium (CM) from ectopic BQ overexpressing cells could induce tamoxifen resistance. MCF-7 and ZR-75 were cultured in CM obtained from MCF-7-BQ and ZR-75-BQ. The original CM was diluted at 0.1× and 0.5×. MCF-7 and ZR-75 cells were treated with 5 µM of tamoxifen (TAM) for 96 h. MTT assay was employed to determine cell viability. (**B**) Conditional medium (CM) from high BQ-expressing cells could induce tamoxifen resistance. MCF-7 and ZR-75 were cultured in CM obtained from LCC2 and AK-47. The original CM was diluted at 0.1× and 0.5×. MCF-7 and ZR-75 cells were treated with 5 µM of TAM for 96 h. MTT assay was performed. (**C**) Treatment of siRNA against IL-6 could reduce the production of IL-6. The cells were treated with the siRNA against IL-6 (siIL-6) or non-targeting siRNA (siCtrl) for 72 h and ELISA was performed on the cell lysates to detect the amount of IL-6. (**D**) CM from siIL-6-treated cells lost the ability to induce tamoxifen resistance. MCF-7 and ZR-75 were cultured in CM from MCF-7-BQ and ZR-75-BQ, which were treated with siCtrl or siIL-6 for 72 h. MCF-7 and ZR-75 cells were treated with 5 µM of TAM for 96 h. MTT assay was performed. (**E**) Knockout of IL-6 could reduce cell viability of BQ-overexpressing cells and resume tamoxifen sensitivity. IL-6 gene was stably knocked out in MCF-7-BQ and ZR-75-BQ. Plasmid-expressing non-targeting gRNA was used as the control. The cells were treated with 5 µM of TAM for 96 h. MTT assay was performed. (**F**) IL-6 could confer tamoxifen resistance. MCF-7 and ZR-75 cells were treated with 10 ng/mL of recombinant IL-6 and 5 µM of TAM for 2 weeks. Colony formation assay was performed to determine cell viability. (**G**) Statistical analysis of (**F**). (**H**) Inhibition of STAT3 could abolish the effect of IL-6 on tamoxifen resistance. One micrometre of STAT3 inhibitor S3I-201, 10 ng/mL of recombinant IL-6, and 5 µM of TAM were used. MTT assay was performed after 96 h of the treatment. Results were shown as mean ± SD from at least three independent experiments. Students’ *t* test was used to determine the statistical significance between two groups. *** represents *p* < 0.001; NS represents no statistical significance.

**Figure 3 cancers-13-01511-f003:**
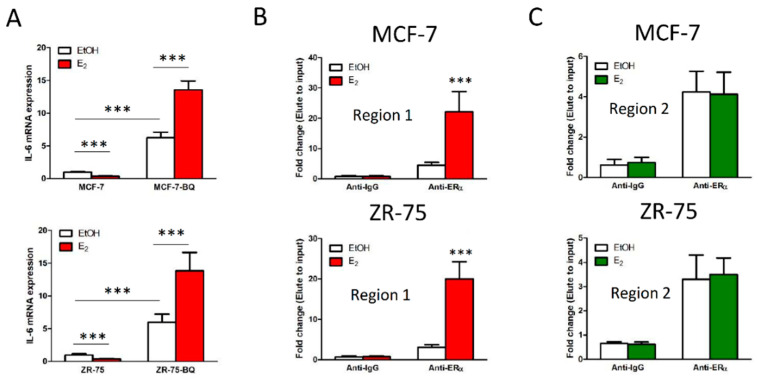
Overexpression of BQ could enhance IL-6 transcription through an Erα-dependent mechanism. (**A**) Overexpression of BQ could alter the effect of estrogen (E_2_) on IL-6 expression. One nanomolar of E_2_ was used. qPCR was performed after 48 h of the treatment. Actin was used as the internal control. Untreated cells were used as the reference. (**B**) E_2_ could enhance ER binding to region 1 of IL-6 promoter. Chromatin immunoprecipitation (ChIP) was performed to determine protein–DNA interaction. ERα was immunoprecipitated. qPCR was used to determine the presence of the target DNA sequence. (**C**) E_2_ could not enhance ERα binding to region 2 of IL-6 promoter. (**D**) Overexpression of BQ could enhance ERα binding to region 1 of IL-6 and the effect was further enhanced in the presence of E_2_. One nanomolar of E_2_ was used. Stably transfected cell lines were used. ChIP assay was performed after 48 h of E_2_ treatment. ERα was immunoprecipitated. qPCR was employed to determine the relative amount of DNA containing region 1 of IL-6 in the immunoprecipitant. (**E**) Overexpression of NCOR2 could compromise the effect of BQ on IL-6 expression in a dose-dependent manner. MCF-7-BQ and ZR-75-BQ cells were transfected with different amounts of *pCMV-NCOR2*. pCMV6 was used as the control overexpression (ctrl OE). qPCR was performed 72 h post transfection to determine the mRNA expression of IL-6. Untransfected MCF-7-BQ or ZR-75-BQ was used as reference. Actin was used as internal control. (**F**) Schematic diagram showing the effect of BQ overexpression on the modulation of IL-6 expression through the ER-dependent pathway. In the presence of BQ overexpression, BQ binds with NCOR2. This interaction compromises the ability of NCOR2 to bind to ERα to repress transcription mediated by ERα. Results were shown as mean ± SD from at least three independent experiments. Students’ *t* test was used to determine the statistical significance between two groups. *, **, and *** represent *p* < 0.05, *p* < 0.01 and *p* < 0.001 respectively.

**Figure 4 cancers-13-01511-f004:**
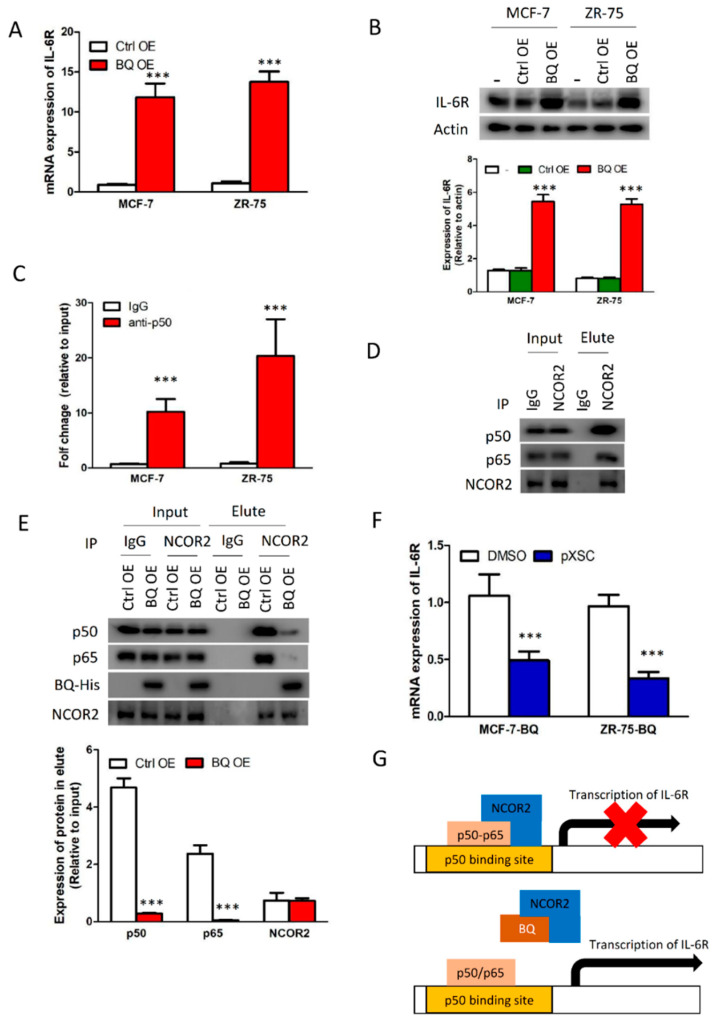
Overexpression of BQ could modulate the expression of IL-6 receptor (IL-6R) in breast cancer. Overexpression of BQ could enhance (**A**) mRNA and (**B**) protein levels of IL-6R. qPCR was performed to determine IL-6R expression in stable BQ-overexpressing cells and their control cells. Parental MCF-7 and ZR-75 were used as reference. Western blot was employed to determine IL-6R. Actin was used as loading control. (**C**) p50 could bind to the promoter of IL-6R. ChIP assay was performed with anti-p50. qPCR was used to detect the relative amount of DNA with the promoter of IL-6R. (**D**) NCOR2 could interact with p50 and p65. p50 and p65 are two subunits of NF-ĸB. Co-immunoprecipitation was performed on MCF-7. Anti-NCOR2 was used to immunoprecipitate the protein complex. Western blot was performed to determine the presence of the indicated protein candidates. (**E**) Overexpression of BQ could compromise the interaction between NCOR2 and NF-ĸB. Co-immunoprecipitation was performed on MCF-7 with anti-NCOR2. Western blot was employed to determine the presence of the indicated proteins in the immunoprecipitant. (**F**) Inhibition of NF-ĸB could suppress the expression of IL-6R in BQ overexpressing cells. Two hundred nanomolar of NF-ĸB inhibitor pXSC was used. The cells were treated for 72 h. qPCR was performed to determine IL-6R expression. Actin was used as internal control. (**G**) Schematic diagram showing the effect of BQ overexpression on the modulation of IL-6R expression through the NF-ĸB-dependent pathway. In the presence of BQ overexpression, BQ binds with NCOR2. This interaction compromises the ability of NCOR2 to bind to NF-ĸB to repress transcription mediated by NF-ĸB. Results were shown as mean ± SD from at least three independent experiments. Students’ *t* test was used to determine the statistical significance between two groups. *** represents *p* < 0.001.

**Figure 5 cancers-13-01511-f005:**
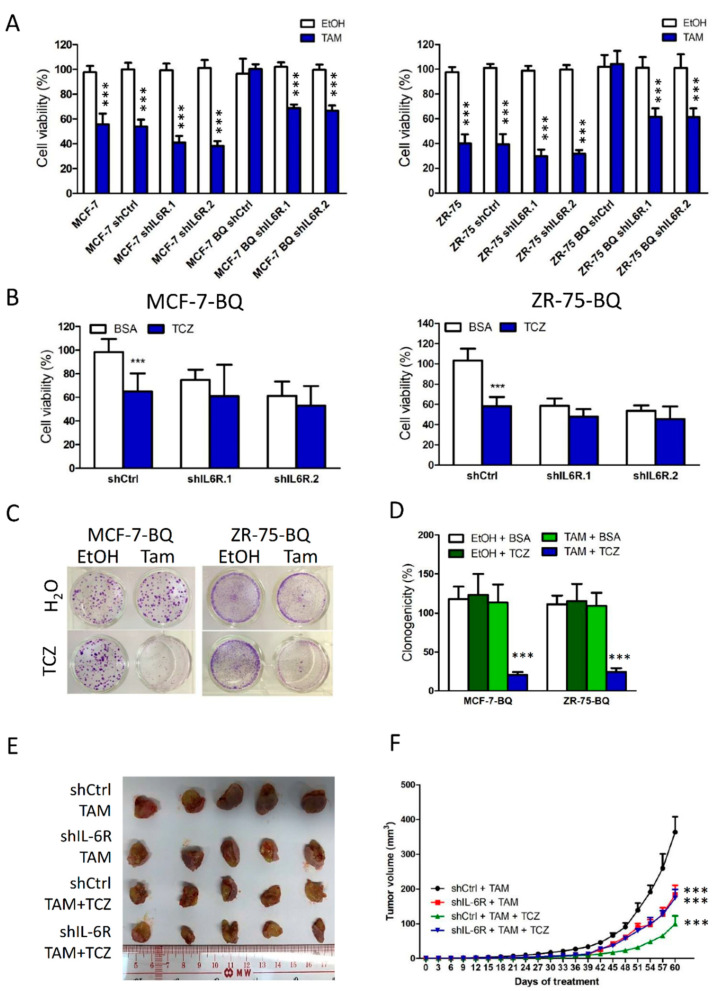
Targeting IL-6R could reverse tamoxifen (TAM) resistance in BQ overexpressing ER+ breast cancer. (**A**) Knockdown of IL-6R could enhance the efficacy of TAM in control cells and reverse TAM resistance in BQ overexpressing cells. Cell lines with stable IL-6R knockdown were used. The cells were treated with 5 µM of TAM for 96 h. MTT was used to determine cell viability. (**B**) Tocilizumab (TCZ) could reduce cell viability in BQ overexpressing cells in the presence of 5 µM of TAM. The effect of TCZ on cell viability was compromised by IL-6R knockdown. Two-hundred-and-fifty ng/mL of TCZ or BSA was used. MTT assay was performed after 96 h of the treatment. (**C**) TCZ could reverse TAM resistance in BQ-overexpressing cells. The cells were treated with 250 ng/mL of TCZ and 5 µM of TAM for 2 weeks. Clonogenic assay was performed. (**D**) Statistical analysis of (**C**). (**E**) TCZ could reduce TAM resistant in vivo. TAM-resistance cell lines LCC2 shCtrl and LCC2 shIL-6R (shIL-6R.1) were used for xenograft establishment. The mice were randomized into different groups. The mice received 0.5 mg of TAM and 2 mg/Kg of TCZ through subcutaneous injection. The mice were treated twice per week for 8 weeks. (**F**) The graph showed the volume change of tumors during the treatment period. Results were shown as mean ± SD from at least three independent experiments. Students’ *t* test was used to determine the statistical significance between treatment and control groups. *** represents *p* < 0.001.

**Figure 6 cancers-13-01511-f006:**
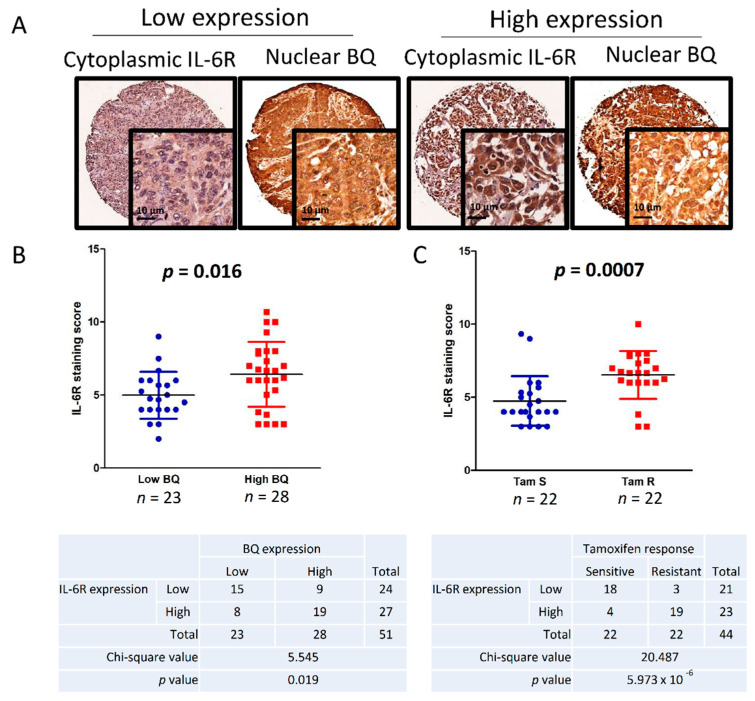
Clinical significance of IL-6R in ER+ breast cancer. (**A**) IHC showing representative high and low expressions of cytoplasmic IL-6R and nuclear BQ respectively. (**B**) Expression of IL-6R was compared with low nuclear BQ and high nuclear BQ expression groups. Mann–Whitney U test was employed to determine the statistical significance between two groups. Chi-square text was used to determine the correlation between the expression of IL-6R and BQ. (**C**) Expression of IL-6R was significantly high (*p* = 0.005; Mann–Whitney U test; *p* = 1.9 × 10^−5^; Chi-square test). Kaplan–Meier analysis showing (**D**) overall survival (*p* = 0.002; log-rank test) and (**E**) disease-specific survival (*p* = 0.003; log-rank test). Patients with high expression of IL-6R (*n* = 29) have a poorer survival outcome compared with patients with low IL-6R expression (*n* = 27).

**Table 1 cancers-13-01511-t001:** Clinical characterization of all breast cancer patients in the current study.

Clinical Characteristic	Type	Cases	Percentage (%)
Number of breast cancer patients		132	
Median Age		56	
T stage			
	I	16	12.1
	II	25	18.9
	III	5	3.8
	Missing	86	65.2
Lymph Node status			
	Positive	62	47.0
	Negative	56	42.4
	Missing	14	10.6
Tumor Grade			
	1	19	14.4
	2	29	22.0
	3	71	53.8
	Missing	13	9.8
Tumor Size			
	<2 cm	37	28.0
	≥2 cm	53	40.2
	Missing	42	31.8
Estrogen Receptor status			
	Positive	71	53.8
	Negative	23	17.4
	Missing	38	28.8
Progesterone receptor status			
	Positive	48	36.3
	Negative	34	25.8
	Missing	50	37.9
HER2 receptor status			
	Positive	33	25.0
	Negative	34	27.3
	Missing	50	47.7
Triple Negative status			
	Positive	12	9.10
	Negative	68	51.5
	Missing	52	39.4

**Table 2 cancers-13-01511-t002:** Clinical characterization of ER+ breast cancer patients in the current study.

Clinical Characteristic	Type	Cases	Percentage (%)
Number of ER+ breast cancer patients		71	
Median Age		51	
T stage			
	I	8	11.3
	II	18	25.3
	III	3	4.2
	Missing	42	59.2
Lymph Node status			
	Positive	37	52.1
	Negative	28	39.4
	Missing	6	8.5
Tumor Grade			
	1	13	18.3
	2	21	29.6
	3	36	50.7
	Missing	1	1.4
Tumor Size			
	<2 cm	22	31.0
	≥2 cm	32	45.1
	Missing	17	23.9
Progesterone receptor status			
	Positive	47	66.2
	Negative	12	16.9
	Missing	12	16.9
HER2 receptor status			
	Positive	27	38.0
	Negative	23	32.4
	Missing	21	29.6

**Table 3 cancers-13-01511-t003:** Cox regression analyses of overall survival in ER+ breast cancer patients.

Clinical-Pathological Parameters	Univariate Analysis		Multivariate Analysis	
Clinical characteristic	RR (95% CI)	*p* Value	RR (95% CI)	*p* Value
Age (*n* = 69)	1.975 (0.931, 4.188)	0.076		
T-stage (*n* = 28)	8.097 (1.441, 45.491)	0.018	4.262 (0.561, 32.405)	0.161
Lymph-node involvement (*n* = 63)	0.904 (0.412, 1.986)	0.802		
Tumor-Grade (*n* = 68)	1.171 (0.550, 2.497)	0.682		
Histological type (*n* = 69)	1.166 (0.351, 3.873)	0.802		
HER2 status (*n* = 48)	1.159 (0.445, 3.016)	0.762		
Tumor size (*n* = 52)	0.941 (0.388, 2.278)	0.892		
Cases with high IL-6R cytoplasm score (*n* = 55)	3.716 (1.537, 8.984)	0.004	10.967 (1.169, 102.878)	0.036

**Table 4 cancers-13-01511-t004:** Cox regression analyses of disease specific survival in ER+ breast cancer patients.

Clinical-Pathological Parameters	Univariate Analysis		Multivariate Analysis	
Clinical characteristic	RR (95% CI)	*p* Value	RR (95% CI)	*p* Value
Age (*n* = 69)	1.198 (0.472, 3.040)	0.703		
T-stage (*n* = 28)	8.097 (1.441, 45.491)	0.018		
Lymph-node involvement (*n* = 63)	1.402 (0.508, 3.864)	0.514		
Tumor-Grade (*n* = 68)	3.672 (1.208, 11.162)	0.022	4.612 (1.298, 16.386)	0.018
Histological type (*n* = 71)	1.022 (0.235, 4.452)	0.976		
HER2 status (*n* = 48)	1.777 (0.517, 6.110)	0.361		
Tumor size (*n* = 52)	1.472 (0.442, 4.898)	0.529		
Cases with high IL-6R cytoplasm score (*n* = 55)	5.664 (1.569, 20.441)	0.008	5.586 (1.534, 20.349)	0.009

## Data Availability

Data is contained within the Appendix A.

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
