# Peer review of "Targeting the IL-6/STAT3 Signalling Cascade to Reverse Tamoxifen Resistance in Estrogen Receptor Positive Breast Cancer"

_cancers, 2021, doi:10.3390/cancers13071511_

Round 1

Reviewer 1 Report

This manuscript presents an interesting study regarding the role of the IL-6/STAT3 signaling pathway in tamoxifen resistance of ER-positive breast cancer. The authors showed that BQ323636.1, a splice variant of NCOR2, modulates the expression of IL-6 and IL-6R and thus the activation of STAT3 in breast cancer in vitro and in vivo. The authors also showed that targeting the IL-6/STAT3 signaling pathway by tocilizumab reversed tamoxifen resistance. Moreover, they also demonstrated an association between high IL-6R expression and poor overall survival in ER-positive breast cancer. The study is well organized, and novel and important findings are shown. However, there are several concerns about the analysis of the clinical samples.

Major comments:

  1. “Stage IV” is included in Table 1A. Patients with Stage IV should be excluded from the prognostic analysis.
  2. In Figure 6E, is the term “disease-free survival” correct? The term “disease specific survival” is not the same as “disease-free survival”.
  3. How did the authors estimate expression of ER, PR and HER2 in breast tissue samples? Number of samples were not analyzed HER2 status (Table 1). Analysis of ER, PR and HER2 status by immunohistochemistry (IHC) using tissue microarray (TMA) blocks is recommended.
  4. The authors showed that high IL-26 6R expression remained significantly associated with poor prognosis in ER-positive breast cancer. Is Stat3 constitutively activated in high IL-6R cells in breast cancer tissues? Expression of phospho-Stat3 by IHC using TMA blocks is strongly recommended.

Minor comments:

  1. Page 1, line 8: “BQ” should be changed to “BQ323636.1”.

Author Response

This manuscript presents an interesting study regarding the role of the IL-6/STAT3 signaling pathway in tamoxifen resistance of ER-positive breast cancer. The authors showed that BQ323636.1, a splice variant of NCOR2, modulates the expression of IL-6 and IL-6R and thus the activation of STAT3 in breast cancer in vitro and in vivo. The authors also showed that targeting the IL-6/STAT3 signaling pathway by tocilizumab reversed tamoxifen resistance. Moreover, they
also demonstrated an association between high IL-6R expression and poor overall survival in ERpositive breast cancer. The study is well organized, and novel and important findings are shown. However, there are several concerns about the analysis of the clinical samples.

Major comments:
1. “Stage IV” is included in Table 1A. Patients with Stage IV should be excluded from the prognostic analysis.

Response:
Thank you for your suggestion. We have excluded the two patients with stage IV in our cox regression analysis and KM survival analysis.

2. In Figure 6E, is the term “disease-free survival” correct? The term “disease specific survival” is not the same as “disease-free survival”.

Response:
Thank you for your comment. Indeed, it is our mistake. The label in Figure 6E should be diseasespecific survival. Thank you very much for reminding us.

3. How did the authors estimate expression of ER, PR and HER2 in breast tissue samples?
Number of samples were not analyzed HER2 status (Table 1). Analysis of ER, PR and HER2 status by immunohistochemistry (IHC) using tissue microarray (TMA) blocks is recommended.

Response
Thank you for questions. The status of ER, PR and HER2 were obtained from pathological reports from Queen Mary hospital. ER, PR and HER2 in the primary tumors were examined by IHC and cases with equivocal HER2 expression were examined by FISH.

This information has been added into materials and method section

4. The authors showed that high IL-26 6R expression remained significantly associated with poor prognosis in ER-positive breast cancer. Is Stat3 constitutively activated in high IL-6R cells in breast cancer tissues? Expression of phospho-Stat3 by IHC using TMA blocks is strongly recommended.

Response:
Thank you for your suggestion. We agree that expression of pSTAT3 and STAT3 in breast cancer tissues should be expected to have clinical significance. Previous studies in gastric adenocarcinomas (1) and Osteosarcoma (2), have shown pSTAT3 to be a poor prognostic factor. Therefore, we expect that pSTAT3
will probably be associated with poorer survival in our study cohort. It is however impossible to complete this analysis within the 10 days we are given to respond to reviewers’ comments. Indeed, many research groups have already addressed the clinical significance of pSTAT3 and STAT3 in various cancer types. Given the constraints of time, we therefore focused our efforts on studying the
upstream mediator of STAT3 pathway, performing IHC with IL-6R rather than STAT3 or pSTAT3.

Minor comments:
1. Page 1, line 8: “BQ” should be changed to “BQ323636.1”.

Response:
Thank you for your suggestion. We have made the change.

Reviewer 2 Report

This manuscript by Tsoi et. al uncovered the molecular mechanisms of BQ323636.1 (BQ) in controlling the Tamoxifen resistance in ER-positive breast cancer. BQ can enhance IL-6/STAT3 signaling in ER-positive breast cancer cells while interfering the pathway either by genetic ablation or pharmacological inhibition can sensitize the BQ expressed cells to Tamoxifen treatment. Overall, this present novel finding which will be interesting to the ER-positive breast cancer community and provide a new therapeutic method for the Tamoxifen resistant patient. However, several concerns should be addressed prior to acceptance.

  1. Line 285-288: What is the rationale to choose these 84 cancer-related genes here? do other genes have been examined? In figure 1A, how many common genes show significantly differential expressions in both cells? Gene name and scale bar are hard to see. In addition, does the heatmap indicate the relative expression/ log transferred fold change of the BQ cells vs parental cells?
  2. Line 298-301: Do the siRNAs also target the wild-type NCOR2? Also, one will curious why not exam the pSTAT3 level in figure 1G? Rescue experiments with either expressing siRNA-resistant BQ or treating with IL-6 may address the on-target effect of siRNA, which is mediated by BQ knockdown.
  1. Figure 1B, show the p-value as well as indicates how many genes enriched in each pathway. Figure 1E, the total STAT3 should be blot as a control in western blot, the quantification of pSTAT3 should be relative to total STAT3. Besides, adding IL-6 treatment will be a positive control.
  2. Line 379-380: Since exogenous NCOR2 expression could lead to IL-6 decline in BQ cells, it seems like that the NCOR2-BQ relative expression level determines the IL-6 expression and sensitivity to tamoxifen. Has the NCOR2-BQ expression ratio been examined in different ER cell lines?
  3. Line 412: There is a lack of evidence that the p50/NF-kB pathway positively regulates IL-6R in the ER+ cells, it will be helpful to check the IL-6R expression by depleting p50.
  4. Line 416: from Figure 4E, BQ could be pulled down by NCOR2, how to explain the interaction between BQ and NCOR2? Does p50/p65 bind with NCOR2 in BQ high expressed cell line such as LCC2?
  5. Line 456: It seems like in Figure 5B, cells were not treated by TAM, therefore this data cannot suggest TCZ treatment could reverse TAM resistance.
  6. Line 464-466: Tumor size in Figure 5E does not correlate with the tumor volume shown in Figure 5F. It's hard to tell some difference between groups shCtrl TAM and shIL-6R TAM, while the tumor growth in 5F has significant change (almost 2 folds between the two groups). Also, according to 5F, group shCtrl TAM+TCZ should present the smallest tumor size. What's the time point for harvesting these tumors? What's the mechanism of TCZ to cause the sensitivity to TAM? Does TCZ repress the IL-6/STAT3 pathway by blocking the IL-6/IL-6R binding? If it does, genetic depletion of IL-6R or TCZ blocking should have a similar phenotype, while the combination of both (blue line in 5F) maybe will have the strongest tumor inhibition. The results are puzzling.

Author Response

This manuscript by Tsoi et. al uncovered the molecular mechanisms of BQ323636.1 (BQ) in controlling the Tamoxifen resistance in ER-positive breast cancer. BQ can enhance IL-6/STAT3 signaling in ER-positive breast cancer cells while interfering the pathway either by genetic ablation or pharmacological inhibition can sensitize the BQ expressed cells to Tamoxifen treatment. Overall, this present novel finding which will be interesting to the ER-positive breast
cancer community and provide a new therapeutic method for the Tamoxifen resistant patient.

However, several concerns should be addressed prior to acceptance.

1. Line 285-288: What is the rationale to choose these 84 cancer-related genes here? do other genes have been examined? In figure 1A, how many common genes show significantly differential expressions in both cells? Gene name and scale bar are hard to see. In addition, does the heatmap indicate the relative expression/ log transferred fold change of the BQ cells vs parental cells?

Response:
Thank you for your question and suggestion. We would like to determine the effect of BQ on cancer related mechanisms to address the effect on tamoxifen resistance. Therefore, we employed PCR array rather than RNA-sequencing. We employed RT² Profiler™ PCR Array Human Cancer PathwayFinder™. This is a commercial array designed by Qiagen to determine the expression of the specific genes encoding proteins related to the main hallmarks of cancer, i.e.
cell cycle control, apoptosis and cell senescence, signal transduction molecules and transcription factors, adhesion, angiogenesis, invasion and metastasis. The qPCR assays used in this PCR Arrays were optimized to work under standard conditions, enabling a large number of genes to be assayed simultaneously.

We have added a supplementary table (Table S1) to show the relative expression levels of individual genes in four cell lines tested. Therefore, readers may refer to the table for detailed gene name and expression data. The Heatmap represented relative expression of genes. We compared the expression between BQ overexpressing (MCF-7-BQ/ZR-75-BQ) and non-BQ overexpressing cells (MCF-7/ZR-75). After obtaining the relative expression level, the data was
subjected to log2 transformation for plotting the Heatmap.

2. Line 298-301: Do the siRNAs also target the wild-type NCOR2? Also, one will curious why not exam the pSTAT3 level in figure 1G? Rescue experiments with either expressing siRNA-resistant BQ or treating with IL-6 may address the on-target effect of siRNA, which is mediated by BQ knockdown.

Response:
Thank you for your questions. BQ is a splice variant of NCOR2 in which the entire exon 11 has been spliced out. BQ is formed as a result of exon 10 joining exon 12. Therefore, this exon10-exon-12 junction is specific to BQ but it is absent in NCOR2. The siRNAs were designed to target this junction. Based on our results from qPCR, these siBQ.1 and siBQ.2 would not affect the expression of NCOR2. These results are shown in Figure S1B.

Following your suggestions, we have added western blot results showing pSTAT3 and total STAT3 in Figure 1G to show that treatment of siBQ.1 and siBQ.2 could reduce the expression of pSTAT3. In addition, we have performed rescue experiment. We found that addition of IL-6 could compensate the effect of siBQ.1 and siBQ.2 on STAT3 activity in LCC2 cells according to the luciferase reporter assay. These results have been added to Figure S1C.

3. Figure 1B, show the p-value as well as indicates how many genes enriched in each pathway. Figure 1E, the total STAT3 should be blot as a control in western blot, the quantification of pSTAT3 should be relative to total STAT3. Besides, adding IL-6 treatment will be a positive control.

Response
Thank you for your suggestions, we have now included the genes enriched in each of the pathways in Table S2. We have also added the total STAT3 in Figure 1G and performed the quantification as you suggested.

4. Line 379-380: Since exogenous NCOR2 expression could lead to IL-6 decline in BQ cells, it seems like that the NCOR2-BQ relative expression level determines the IL-6 expression and sensitivity to tamoxifen. Has the NCOR2-BQ expression ratio been examined in different ER cell lines?

Response:
The NCOR2-BQ relative expression has been studied and published in our previous manuscripts (3-5) . However, we had not compared the ratio alongside a panel of breast cancer cell lines. We have now performed qPCR in 4 cell lines to determine the expression of NCOR2 and BQ and calculated the ratio of BQ to NCOR2. The result is now shown in supplementary Figure S1A

5. Line 412: There is a lack of evidence that the p50/NF-kB pathway positively regulates IL- 6R in the ER+ cells, it will be helpful to check the IL-6R expression by depleting p50.

Response:
Thank you for your suggestion. We employed NF-kB inhibitor pXSC. Based in the information from Abcam, this inhibitor actually covalently modifies p50 so that the DNA binding activity of p50 will be abolished and thus also abolishing its mediated transcription activity. We demonstrated in figure 4F the effect of pXSC on IL-6R expression in MCF-7-BQ and ZR-75-BQ cells. We found that the inhibitor could reduce the expression of IL-6R. This result suggestes that inhibition of NF-kB by targeting p50 could suppress the transcription of IL-6R mediated by BQ.

6. Line 416: from Figure 4E, BQ could be pulled down by NCOR2, how to explain the interaction between BQ and NCOR2? Does p50/p65 bind with NCOR2 in BQ high expressed cell line such as LCC2?

Response:
Thank you for your question. NCOR2 is known to homodimerize in an anti-parallel manner (6), serving as a dock for further recruitment of other corepressor proteins. BQ as a truncated splice variant of NCOR2, only retains the N-terminus of NCOR2. Therefore, BQ can interact with NCOR2 through the similar mechanism used to form the NCOR2-NCOR2 homodimer (3). When BQ is
present, it binds to NCOR2, forming a faulty dock to which other corepressor proteins cannot be fully recruited, resulting in a defective corepressor complex. Based on the results from figure 4E, overexpression of BQ could interfere with the interaction between NCOR2 and p50/p65. LCC2 is a high BQ expressing cells. Therefore, this overexpressed BQ in binding to NCOR2 will thus compete
for and compromise the interaction between NCOR2 and other interacting partners such as
p50/p65.

7. Line 456: It seems like in Figure 5B, cells were not treated by TAM, therefore this data cannot suggest TCZ treatment could reverse TAM resistance.

Response:
Thank you for pointing out our mistake. We missed giving the information to indicate that all the cells in Figure 5B were cultured in the medium containing 5 μM of tamoxifen. This information has now been added to the legend of figure 5B. Thank you very much for drawing our attention to this omission.

8. Line 464-466: Tumor size in Figure 5E does not correlate with the tumor volume shown in Figure 5F. It's hard to tell some difference between groups shCtrl TAM and shIL-6R TAM, while the tumor growth in 5F has significant change (almost 2 folds between the two groups). Also, according to 5F, group shCtrl TAM+TCZ should present the smallest tumor size. What's the time point for harvesting these tumors? What's the mechanism of TCZ to cause the sensitivity to TAM? Does TCZ repress the IL-6/STAT3 pathway by blocking the
IL-6/IL-6R binding? If it does, genetic depletion of IL-6R or TCZ blocking should have a similar phenotype, while the combination of both (blue line in 5F) maybe will have the strongest tumor inhibition. The results are puzzling.

Response:
Thank you for your comments. We understand your concerns and agree that the tumor image in Figure 5E is not perfect. We harvested the tumor at day 60. During the tumor isolation process, the tumors were a bit fragile and soft. To avoid damage to the tumors, we could not completely remove the skin and other connective tissues surrounding the tumors as this might introduce too
much distortion to the images taken . We have now included the images showing the mice bearing the tumors before tumors were harvested in Figure S5B. The size of tumors in shIL-6R+TAM and shIL-6R+TAM+TCZ was similar, and shCtrl + TAM + TCZ had the smallest tumours. IL-6/STAT3 signaling pathway is not the only pathway associated with tamoxifen resistance. Our previous studies demonstrated that overexpression of BQ could activate the ER signaling pathway (3) and the NRF2 pathway (4). Down-regulation of IL-6R depletes the target for TCZ and thus the efficacy of TCZ was lost. The knockdown resulted in tumors no longer relying on the IL-6/STAT3 signaling pathway. However, BQ can employ other mechanisms to confer resistance to tamoxifen, explaining in part why tumors from shIL-6R+TAM and shIL-6R+TAM+TCZ groups were similar.
Since TCZ is a specific antibody that blocks IL-6R, TCZ could not affect other pathways associated with tamoxifen resistance. Therefore, the addition of TCZ could not increase the efficacy of tamoxifen and thus could not reduce the tumor size.

References
1. H. Xiong et al., Constitutive activation of STAT3 is predictive of poor prognosis in human gastric cancer. J Mol Med (Berl) 90, 1037-1046 (2012).
2. K. Ryu et al., Activation of signal transducer and activator of transcription 3 (Stat3) pathway in osteosarcoma cells and overexpression of phosphorylated-Stat3 correlates with poor prognosis. J Orthop Res 28, 971-978 (2010).
3. C. Gong et al., BQ323636.1, a Novel Splice Variant to NCOR2, as a Predictor for Tamoxifen- Resistant Breast Cancer. Clin Cancer Res 24, 3681-3691 (2018).
4. M. H. Leung et al., A Splice Variant of NCOR2, BQ323636.1, Confers Chemoresistance in Breast Cancer by Altering the Activity of NRF2. Cancers (Basel) 12 (2020).
5. L. D. Zhang et al., SpliceArray Profiling of Breast Cancer Reveals a Novel Variant of NCOR2/SMRT That Is Associated with Tamoxifen Resistance and Control of ER alpha Transcriptional Activity. Cancer Res 73, 246-255 (2013).
6. N. Varlakhanova, J. B. Hahm, M. L. Privalsky, Regulation of SMRT corepressor dimerization and composition by MAP kinase phosphorylation. Mol Cell Endocrinol 332, 180-188 (2011).

Round 2

Reviewer 1 Report

The authors have revised the manuscript carefully and addressed all of my concerns.

Reviewer 2 Report

The authors have addressed all the concerns from the reviewer. 

Additional comments.

Enlarge the font of some figures to make them readable. Also, the figures are a little blurry but this will probably be fixed in the production step.